# Unpaired Photo-realistic Image Deraining with Energy-informed Diffusion Model

Yuanbo Wen
School of Information Engineering,
Chang'an University
Xi'an, China
wyb@chd.edu.cn

Tao Gao*
School of Data Science and Artificial
Intelligence, Chang'an University
Xi'an, China
gtnwpu@126.com

Ting Chen
School of Information Engineering,
Chang'an University
Xi'an, China
tchen@chd.edu.cn

## Abstract

Existing unpaired image deraining approaches face challenges in accurately capture the distinguishing characteristics between the rainy and clean domains, resulting in residual degradation and color distortion within the reconstructed images. To this end, we propose an energy-informed diffusion model for unpaired photo-realistic image deraining (UPID-EDM). Initially, we delve into the intricate visual-language priors embedded within the contrastive language-image pre-training model (CLIP), and demonstrate that the CLIP priors aid in the discrimination of rainy and clean images. Furthermore, we introduce a dual-consistent energy function (DEF) that retains the rain-irrelevant characteristics while eliminating the rain-relevant features. This energy function is trained by the non-corresponding rainy and clean images. In addition, we employ the rain-relevance discarding energy function (RDEF) and the rain-irrelevance preserving energy function (RPEF) to direct the reverse sampling procedure of a pre-trained diffusion model, effectively removing the rain streaks while preserving the image contents. Extensive experiments demonstrate that our energy-informed model surpasses the existing unpaired learning approaches in terms of both supervised and no-reference metrics.

## CCS Concepts

• **Computing methodologies → Unsupervised learning**; **Reconstruction**; **Computer vision**.

## Keywords

Computer vision, Image deraining, Unpaired learning, Energy-informed model, Diffusion model

**ACM Reference Format:**
Yuanbo Wen, Tao Gao, and Ting Chen. 2024. Unpaired Photo-realistic Image Deraining with Energy-informed Diffusion Model. In *Proceedings of the 32nd ACM International Conference on Multimedia (MM '24), October 28-November 1, 2024, Melbourne, VIC, Australia.* ACM, New York, NY, USA, 10 pages. https://doi.org/10.1145/3664647.3680560

_______________
*Corresponding author

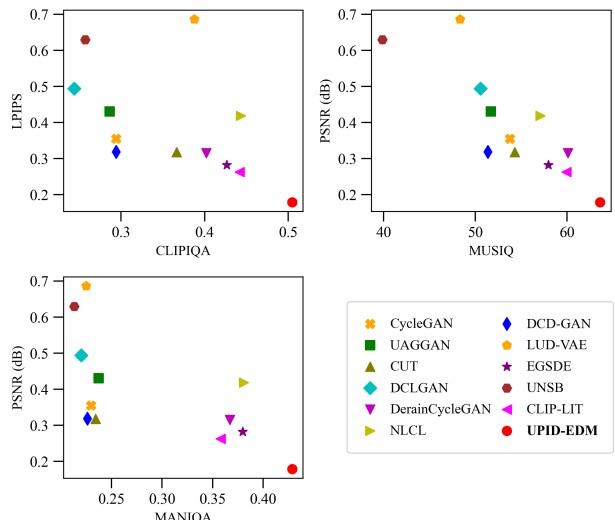

**Figure 1: Intuitive comparisons of our proposed method and the other existing approaches between the average learned perpetual image patch similarity and three image naturalness assessment metrics. Our model achieves the currently best performance in the supervised protocols, while preserving the significant improved naturalness.**

## 1 Introduction

Images taken during rainy weather condition frequently exhibit reduced visibility caused by rain streaks [3, 35, 36, 43]. This degradation severely impedes multiple computer vision tasks [10, 34, 42], including detection, segmentation, and video surveillance. There are significant interests in developing methods to mitigate the rain degradation and reconstruct the photo-realistic details. Despite recent progress in data-driven learning techniques [4, 5, 33, 44, 49, 50], fully-supervised learning that depend on the paired synthetic images frequently fall short in accurately capturing the underlying rain features. Therefore, these approaches encounter difficulties when confronted with real-world rainy images due to the disparity between the datasets utilized for training and testing.

Due to the scarcity of precisely labeled data for image deraining, several strategies have surfaced to tackle this issue. These encompass semi-supervised methodologies, which leverage both labeled and unlabeled data, alongside unpaired learning techniques that function without explicit correlations between rainy and clean images. The former researches [8, 17, 23, 39, 56] prioritize extracting

features that remain consistent across different domains and integrating supplementary objectives to enhance the performance, while unpaired techniques [6, 14, 41, 47, 57] frequently utilize domain adaptation strategies to achieve superior generalization. However, existing methods encounter challenges in attaining superior restoration quality because of the absence of clearly defined constraints for both rainy and clean images, leading to problems like residual degradation and color distortion. Therefore, there is a necessity to create thorough mapping representations that effectively capture the inherent relationship between rainy and clean domains. Fortunately, score-based diffusion model (SBDM) [15, 30] presents a promising departure from traditional generative adversarial network (GAN) [11]. SBDM perturbs data through a diffusion process and master the reversal of this process, yielding image generation performance that is either competitive or superior. However, to the best of our knowledge, there has been no the exploration of utilizing the score-based diffusion model within the unpaired image deraining.

In this work, we propose an energy-informed diffusion model for unpaired photo-realistic image deraining, identified as UPID-EDM. This approach utilizes a dual-consistent energy function (DEF) that has been pre-trained across both rainy and clean domains. It directs the reverse sampling procedure of a pre-trained stochastic differential equation (SDE), facilitating the removal of rain streaks and the restoration of image details. Image deraining can be conceptualized as comprising two distinct processes, namely preserving the rain-irrelevant features (image content) and discarding the rain-relevant features (rain streaks). Based on this, we decompose the energy function into the summation of two potential functions. Although the contrastive language-image pre-training model (CLIP) [26] possesses the capability to discern between images depicting rain and those depicting cleanliness, its practical implementation still presents considerable challenges. For instance, although *clean images* and *rain-free images* are related concepts, we notice varying CLIP scores when comparing with the same given images [21]. To this end, we introduce the trainable domain-representation prompts (LDP), initialized randomly and trained using non-correspondence images. Specifically, we employ the rainy and clean images as inputs for a fixed image encoder, while the learnable negative and positive prompts are utilized as inputs for a fixed text encoder. Subsequently, we adopt the binary cross-entropy loss to classify the rainy and clean images, aiding in the acquisition of learned domain-representation prompts. Furthermore, we establish a rain-relevance discarding energy function (RDEF) to ascertain whether a given image belongs to the clean domain. Meanwhile, we also formulate a rain-irrelevance preserving energy function (RPEF) that ensures the generated images maintain consistency with the given rainy images in terms of image content. With the guidance of our suggested dual-consistent energy function, the pre-trained stochastic differential equation applied to the clean domain can produce the photo-realistic derained images when only given the rainy images.

Figure 2 illustrates the overall pipeline of our proposed approach. The main contributions can be summarized as follows.

- Our work pioneers the utilization of diffusion models in unpaired image deraining, showcasing the potential for the first time.

- We decompose the energy function into two components, aiming to retain rain-irrelevant features while eliminating rain-relevant features during the reverse sampling procedure for generating reconstructed images.
- By leveraging the perceptual abilities of contrastive language-image pre-training model, we propose the learnable domain-representation prompts that guarantee the generated images adhere to clean domain.
- Experiments on publicly available datasets demonstrate our proposed approach achieves the best photo-realistic unpaired deraining performance.

## 2 Related Work

### 2.1 Unpaired Image Deraining

In the realm of unpaired image deraining, generative adversarial network [58] has emerged as a widely favored model. Several recent works [14, 57] improve CycleGAN [58] by integrating constraints tailored for transfer learning, with a particular focus on rainy and clean images. For instance, Wei *et. al.* [40] leverage the unpaired training datasets to tackle the unpaired deraining task, indicating a notable progression. Zhu *et. al.* [57] represent a pioneering instance of an end-to-end adversarial model that solely depends on the unpaired supervision to generate the authentic images. Meanwhile, Yu *et. al.* [48] incorporate the existing data on rain streaks by combining both model-driven and data-driven methods within an unsupervised framework. Recently, Chen *et. al.* [6] combine contrastive learning with adversarial training to bolster the robustness of unpaired deraining methods, while Chang *et. al.* [3] investigate the inherent similarities within each layer and the distinctiveness between two layers, then propose an unsupervised non-local contrastive learning approach for removing the rain effects. However, these approaches primarily either depend on enforcing cycle-consistency constraints, or alternatively, they are based on adversarial training principles, leading to constraints on the overall realism and fidelity of the reconstructed images.

### 2.2 Score-based Diffusion Model

Score-based diffusion model [22, 30, 31] have recently made significant progress in a series of conditional image generation tasks. SBDM provides a diffusion model to guide how image shaped data from a Gauss distribution is iterated step by step into an image of the target domain. In each step, SBDM gives score guidance which, from an engineering perspective, can be mixed with energy and statistical guidance to control the generation process. Saharia *et. al.* [27, 28] develop a conditional SBDM to achieve paired super-resolution and colorization. Choi *et. al.* [7] employ the reference image to refine the generated images after each denoising step with a low-pass filter. Recently, Zhao *et. al.*[54] and Sun *et. al.* [32] decompose the score into several components to guide the reverse sampling process and achieve the competitive image translation performance. The characteristic that SBDM can be influenced by energy or statistics serves as a strong motivation for us to employ the domain representation approaches in order to effectively transition from rainy domain to clean domain. However, there is currently no approach that employ SBDM to achieve unpaired image deraining.

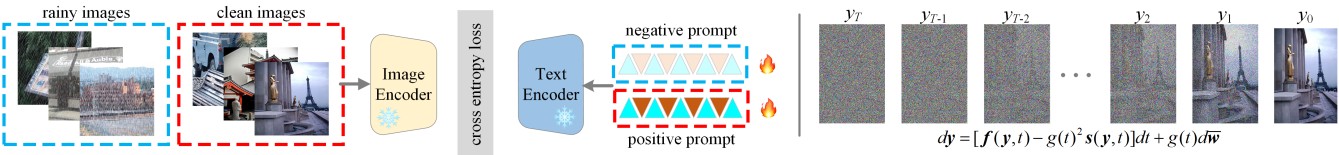

(a) Training of the learnable domain-representation prompts on unpaired image pairs and the diffusion model on clean images.

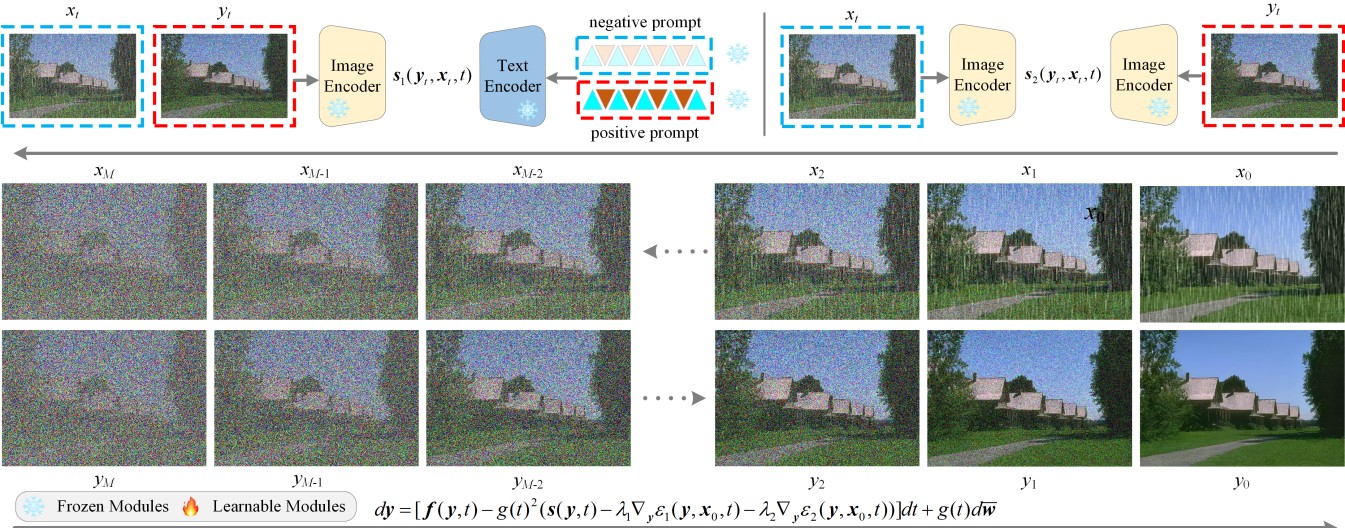

(b) Inference of the pre-trained diffusion model guided by the dual-consistent energy function.

**Figure 2: Overall pipeline of our proposed energy-informed diffusion model for unpaired photo-realistic image deraining (UPID-EDM). This approach employs our developed dual-consistent energy function (DEF) pre-trained on the unpaired rainy and clean images to guide the reverse sampling process of a pre-trained diffusion model. We decompose the energy function into two components, which discard the rain-relevant features and preserve the rain-irrelevant features, respectively.**

## 3  Preliminary

SBDM [31, 53] initially perturb the training data through a forward diffusion process and subsequently acquire the capability to invert this process, thereby constructing a generative model that approximates the underlying data distribution. Let $q(\boldsymbol{y}_0)$ be the unknown data distribution on $\mathbb{R}^D$. The forward diffusion process $\boldsymbol{y}_t$, indexed by time $t$, can be represented by the following forward SDE

$$dy = f(\boldsymbol{y}, t)dt + g(t)d\boldsymbol{w}, \tag{1}$$

where $t \in [0, T]$, $\boldsymbol{w} \in \mathbb{R}^D$ is a standard Wiener process, $\boldsymbol{f}(\cdot, t) : \mathbb{R}^D \rightarrow \mathbb{R}^D$ is the drift coefficient and $g(t) \in \mathbb{R}$ is the diffusion coefficient. Denote $q_{t|0}(\boldsymbol{y}_t|\boldsymbol{y}_0)$ the transition kernel from time 0 to $t$, which is decided by $\boldsymbol{f}(\boldsymbol{y}, t)$ and $g(t)$. In practice, $\boldsymbol{f}(\boldsymbol{y}, t)$ typically exhibits an affine behavior, resulting in the perturbation kernel being a linear Gaussian distribution, allowing for straightforward sampling in a single step. Let $q_t(\boldsymbol{y})$ be the marginal distribution of the SDE at time t in Eq. (1), its time reversal can be described by

$$d\boldsymbol{y} = [f(\boldsymbol{y}, t) - g(t)^2 \nabla_{\boldsymbol{y}} \log q_t(\boldsymbol{y})]dt + g(t)d\bar{\boldsymbol{w}}, \tag{2}$$

where $\bar{w}$ is a reverse-time standard Wiener process, and $dt$ is an infinitesimal negative timestep. Song *et. al.* [31] adopts a score-based model $\boldsymbol{s}(\boldsymbol{y}, t)$ to approximate the unknown $\nabla_{\boldsymbol{y}} \log q_t(\boldsymbol{y})$ by score matching, thereby inducing a score-based diffusion model,

which is defined as

$$dy = [f(\boldsymbol{y}, t) - g(t)^2 s(\boldsymbol{y}, t)]dt + g(t)d\bar{w}. \tag{3}$$

## 4  Proposed Method

### 4.1  Overview

Based on Eq. (3), incorporating a supplementary guidance function $\epsilon(\boldsymbol{y}, \boldsymbol{x}_0, t)$ proves beneficial, leading to the derivation of a revised time-reversed SDE following

$$d\boldsymbol{y} = [f(\boldsymbol{y}, t) - g(t)^2(s(\boldsymbol{y}, t) - \nabla_{\boldsymbol{y}}\epsilon(\boldsymbol{y}, \boldsymbol{x}_0, t))]dt + g(t)d\bar{\boldsymbol{w}}, \tag{4}$$

where $\boldsymbol{s}(\cdot, \cdot) : \mathbb{R}^D \times \mathbb{R} \rightarrow \mathbb{R}^D$ is the score-based diffusion model in the pre-trained SDE and $\epsilon(\cdot, \cdot, \cdot) : \mathbb{R}^D \times \mathbb{R}^D \times \mathbb{R} \rightarrow \mathbb{R}$ is the proposed dual-consistent energy function. The start point $y_{T_s}$ is sampled from the perturbation distribution $q_{T_s|0}(y_{T_s}|x_0)$, where $T_s = 0.4T$ empirically. We acquire the derained images by sampling at the time point $t = 0$ according to the stochastic differential equation outlined in Eq. (4). Meanwhile, our methodology utilizes a score-based diffusion model exclusively trained on the pristine domain. This model delineates a marginal distribution of clean images and predominantly enhances the authenticity of the derained samples. The suggested dual-consistent energy function incorporates data from both rainy and clean domains in an unpaired setting, aiming to maintain rain-irrelevant features while discarding rain-relevant

ones. This approach enhances both the fidelity and naturalness of the reconstructed images. After individually completing training, we employ Eq. (13) to sample the reconstructed images based on the given rainy images.

## 4.2 Dual-consistent Energy Function

Image deraining can be described as a dual process involving the preservation of rain-irrelevant features (image content) and the removal of rain-relevant features (rain streaks). Therefore, heavily based on [54], we express the energy function $\epsilon(\boldsymbol{y}, \boldsymbol{x}, t)$ as the combination of two logarithmic potential functions,

$$
\begin{aligned}
\epsilon(\boldsymbol{y}, \boldsymbol{x}, t) &= \lambda_1 \epsilon_1(\boldsymbol{y}, \boldsymbol{x}, t) + \lambda_2 \epsilon_2(\boldsymbol{y}, \boldsymbol{x}, t) \\
&= \lambda_1 \mathbb{E}_{q_{t|0}(\boldsymbol{x}_t|\boldsymbol{x})} s_1(\boldsymbol{y}, \boldsymbol{x}_t, t) + \lambda_2 \mathbb{E}_{q_{t|0}(\boldsymbol{x}_t|\boldsymbol{x})} s_2(\boldsymbol{y}, \boldsymbol{x}_t, t),
\end{aligned} \tag{5}
$$

where $\epsilon_1(\cdot, \cdot, \cdot) : \mathbb{R}^D \times \mathbb{R}^D \times \mathbb{R} \to \mathbb{R}$ and $\epsilon_2(\cdot, \cdot, \cdot) : \mathbb{R}^D \times \mathbb{R}^D \times \mathbb{R} \to \mathbb{R}$ are the log potential functions, $x_t$ is the perturbed source images in the forward diffusion process, $q_{t|0}(\cdot|\cdot)$ is the perturbation kernel from time 0 to time $t$ in the forward diffusion process, $s_1(\cdot, \cdot, \cdot) : \mathbb{R}^D \times \mathbb{R}^D \times \mathbb{R} \to \mathbb{R}$ and $s_2(\cdot, \cdot, \cdot) : \mathbb{R}^D \times \mathbb{R}^D \times \mathbb{R} \to \mathbb{R}$ are two functions measuring the similarity between the sample and perturbed source image, and $\lambda_1 \in \mathbb{R} > 0$, $\lambda_2 \in \mathbb{R} > 0$ are two weighting hyperparameters.

To specify $s_1(\cdot, \cdot, \cdot)$, we investigate the diverse priors within the contrastive language-image pre-training model, which exhibit a robust capability to capture domain discrimination [21, 45]. However, we discover that texts conveying similar meanings exhibit considerable separation within the latent space. As shown in Figure 5, we present two instances featuring a single image alongside varied but analogous texts, followed by an assessment of their similarities within the CLIP space. Therefore, we employ the learnable domain-representation prompts to depict the rainy and clean domains without the need for paired images. Based on the inherent latent resemblance between images and prompts, we employ binary cross-entropy loss to classify images as either rainy or clean. This loss function is derived from

$$
\mathcal{L}_{prompt} = -[z \log(\hat{z}(\boldsymbol{v}, \boldsymbol{p}_p)) + (1-z) \log(1 - \hat{z}(\boldsymbol{v}, \boldsymbol{p}_p))], \tag{6}
$$

where $\hat{z}(\boldsymbol{v}, \boldsymbol{p}) = p(z|\mathcal{E}_i(\boldsymbol{v})) \in \{0, 1\}$ denotes the target label and prediction probability, $\boldsymbol{v} \in \{\boldsymbol{x}_t, \boldsymbol{y}\}$, $p(\cdot)$ indicates probability, $\boldsymbol{p}_p$ is the learnable positive prompt. $y = 0$ is the label of rainy images and $y = 1$ is the label of clean images. In our setting, the original prediction probability can be formulated as

$$
\hat{z}(\boldsymbol{v}, \boldsymbol{p}_p) = \frac{exp(sim(\mathcal{E}_i(\boldsymbol{v}), \mathcal{E}_t(\boldsymbol{p}_p)))}{\sum_{u \in \{n,p\}} exp(sim(\mathcal{E}_i(\boldsymbol{v}), \mathcal{E}_t(\boldsymbol{p}_u)))}, \tag{7}
$$

where $sim(\cdot)$ denotes the cosine similarity, $\mathcal{E}_i(\cdot)$ and $\mathcal{E}_t(\cdot)$ indicate the image encoder and text encoder in CLIP, respectively. $\boldsymbol{p}_n$ is the learnable negative prompts. As illustrated in Figure 5, our learnable domain-representation prompts facilitate the accuracy of CLIP in distinguishing between rainy and clean images.

Following the training of the learnable prompts, we also select the negative prompt to represent the underlying correlation between the provided images and prompts [21], a formulation of which can be described as

$$
\hat{z}(\boldsymbol{v}, \boldsymbol{p}_n) = \frac{exp(sim(\mathcal{E}_i(\boldsymbol{v}), \mathcal{E}_t(\boldsymbol{p}_n)))}{\sum_{u \in \{n,p\}} exp(sim(\mathcal{E}_i(\boldsymbol{v}), \mathcal{E}_t(\boldsymbol{p}_u)))}. \tag{8}
$$

---

**Algorithm 1** Energy-informed Diffusion Model

---

**Input:** rainy image $\boldsymbol{x}_0$, initial time $T_s$, denoising steps $N$, weighting hyper-parameters $\lambda_1, \lambda_2$, dual-consistent energy function $s_1(\cdot, \cdot, \cdot)$ and $s_2(\cdot, \cdot, \cdot)$, score function $\boldsymbol{s}(\cdot, \cdot)$

**Output:** reconstructed image $y_0$

1: $\boldsymbol{y}_{T_s} \sim q_{M|0}(\boldsymbol{y}_{T_s}|\boldsymbol{x}_0)$
2: $h = \frac{T_s}{N}$
3: **for** $i = N$ to $1$ **do**
4: $\quad n = ih$
5: $\quad t = n - h$
6: $\quad \boldsymbol{x} \sim q_{n|0}(\boldsymbol{x}|\boldsymbol{x}_0)$
7: $\quad \epsilon(\boldsymbol{y}_n, \boldsymbol{x}_0, n) = \lambda_1 \epsilon_1(\boldsymbol{y}_n, \boldsymbol{x}_0, n) - \lambda_2 \epsilon_2(\boldsymbol{y}_n, \boldsymbol{x}_0, n)$
8: $\quad \boldsymbol{y}_t = \boldsymbol{y}_n - [f(\boldsymbol{y}_n, n) - g(n)^2(\boldsymbol{s}(\boldsymbol{y}_n, n) - \nabla_{\boldsymbol{y}}\epsilon(\boldsymbol{y}_n, \boldsymbol{x}_0, n))]h + g(n)\sqrt{h}\eta, \eta \sim \mathcal{N}(\boldsymbol{0}, \boldsymbol{I})$ if $i > 1$, else $\eta = 0$
9: **end for**
10: **return** $\boldsymbol{y}_0$

---

Therefore, we have the rain-relevance discarding energy function $s_1(\boldsymbol{y}, \boldsymbol{x}_t, t)$ following

$$
s_1(\boldsymbol{y}, \boldsymbol{x}_t, t) = \hat{z}(\boldsymbol{x}_t, \boldsymbol{p}_n) + \hat{z}(\boldsymbol{y}, \boldsymbol{p}_n). \tag{9}
$$

Lowering this energy value indicates the generated sample to eliminate rain-relevant characteristics, ensuring its alignment with the clean domain [2, 12].

Furthermore, to maintain rain-irrelevant attributes and ensure contextual coherence between generated samples and clean images, we utilize two CLIP image encoders to assess latent similarity. This approach aims to enhance the alignment of derained images with their rainy counterparts in terms of contextual features [21], thereby fostering the fidelity of content within the derained images. Therefore, we formulate the rain-irrelevance preserving energy function $s_2(\boldsymbol{y}, \boldsymbol{x}_t, t)$ as

$$
s_2(\boldsymbol{y}, \boldsymbol{x}_t, t) = \frac{1}{m} \sum_{k=0}^{m} \lambda_k ||\mathcal{E}_i^k(\boldsymbol{y}) - \mathcal{E}_i^k(\boldsymbol{x}_t)||_2, \tag{10}
$$

where $m$ denotes the number of image encoder layers utilized to represent the distances, $\mathcal{E}_i(\cdot)$ is the weight of $k$-th layer in image encoder, $\lambda_k$ indicates the coefficient corresponding to the $k$-th layer. By decreasing this energy value, it prompts the improved outcome to closely resemble the clean images in content, thereby safeguarding the rain-irrelevant features. Therefore, our decomposed dual-consistent energy function aids our model in filtering out rain-relevant features while retaining those that are rain-irrelevant. Compared to the energy function in [54], our decomposed energy function achieves better domain representation and structural preservation in the complex generative process of image deraining.

## 4.3 Energy-informed diffusion model

In this work, we utilize the Euler-Maruyama solver [24] to tackle the energy-guided reverse-time stochastic differential equation. Based on the pre-trained score-based model $\boldsymbol{s}(\boldsymbol{y}, t)$ and energy function $\epsilon(\boldsymbol{y}, \boldsymbol{x}, t)$, we can solve the proposed energy-informed score-based diffusion model to generate samples from conditional distribution

$p(\boldsymbol{y}_0|\boldsymbol{x}_0)$. We select a step size $h$, and utilize the iteration rule starting from $t = n - h$ following

$$
\begin{aligned}
\boldsymbol{y}_t =& \boldsymbol{y}_n - [f(\boldsymbol{y}_n, n) - g(n)^2(s(\boldsymbol{y}_n, n) - \nabla_{\boldsymbol{y}}\epsilon(\boldsymbol{y}_n, \boldsymbol{x}_0, n))]h \\
&+ g(n)\sqrt{h}\eta,
\end{aligned}
\tag{11}
$$

where $\eta \sim \mathcal{N}(\boldsymbol{0}, \boldsymbol{I})$. The anticipated value within $\epsilon(\boldsymbol{y}_s, \boldsymbol{x}_0, s)$ is evaluated using the Monte Carlo method with a single sample to enhance efficiency. We outline the overarching sampling process of our approach in Algorithm 1. In our experiments, we employ the variance-preserving energy-informed diffusion model, as explicitly defined by

$$
d\boldsymbol{y} = [-\frac{1}{2}\beta(t)\boldsymbol{y} - \beta(t)(s(\boldsymbol{y}, t) - \nabla_{\boldsymbol{y}}\epsilon(\boldsymbol{y}, \boldsymbol{x}_0, t))]dt + \sqrt{\beta(t)}d\bar{w}, \tag{12}
$$

where $\beta(\cdot)$ is a positive function. The perturbation kernel $q_{t|0}(\boldsymbol{y}_t|\boldsymbol{y}_0) \sim \mathcal{N}(\boldsymbol{y}_0 \exp{-\frac{1}{2}\int_0^t \beta(n)dn}, (1-\exp{-\int_0^t (\beta(n)dn)}\boldsymbol{I})$ and $\beta(t) = \beta_{min} + t(\beta_{max} - \beta_{min})$. We set $\beta_{min} = 0.1$ and $\beta_{max} = 20$ [15, 24, 54]. Therefore, the iteration rule from $n$ to $t = n - h$ is

$$
\begin{aligned}
\boldsymbol{y}_t =& \frac{1}{\sqrt{1 - \beta(n)h}}[\boldsymbol{y}_s + \beta(n)h(s(\boldsymbol{y}_n, n) - \nabla_{\boldsymbol{y}}\epsilon(\boldsymbol{y}_n, \boldsymbol{x}_0, n))] \\
&+ \sqrt{\beta(n)h\eta}.
\end{aligned}
\tag{13}
$$

The iteration rule in Eq. (13) is equivalent to that using Euler-Maruyama solver when $h$ is small [31], thereby the score network is modified to $\tilde{s}(\boldsymbol{y}, \boldsymbol{x}_0, t) = s(\boldsymbol{y}, t) - \nabla_{\boldsymbol{y}}\epsilon(\boldsymbol{y}, \boldsymbol{x}_0, t)$ in our proposed UPID-EDM. Therefore, we can readily adjust the noise prediction network and integrate it into the reverse sampling process [15].

## 5 Experimental Results

### 5.1 Implementation Details

We implement our proposed method with PyTorch on a single NVIDIA GeForce RTX 4090 GPU. The rain-relevance discarding energy function is trained on the combined dataset, where we ensure the rainy images are never paired with their corresponding clean images. We utilize the Adam optimizer [20] with $\beta_1 = 0.9$ and $\beta_2 = 0.99$. Each learnable domain-representation prompt contains 16 embedded tokens, with a total of 50 000 iterations. The learning rate for the energy function learning is set to $5 \times 10^{-6}$, and the batch size is 8. In the rain-irrelevance preserving energy function, $\lambda_{k \in 1,2,3,4,5} = 0.5, 1, 1, 1, 1$. The initial time $T_s$ is configured as $0.4T$.

### 5.2 Datasets

We employ five commonly recognized benchmark rainy datasets to assess the effectiveness of our proposed approach, namely Rain800 [52], Rain1400 [9], Rain1200 [51], RainCityscapes [16], SPA-Data [38] datasets. In this work, our training regimen integrates a combination of these datasets, while testing experiments are conducted independently. A detailed description of these rainy datasets is provided in Table 1. To optimize our training, we utilize a feature clustering approach to eliminate data that deviates significantly from the center of the testing distribution.

### 5.3 Evaluation Metrics

We employ the learned perpetual image patch similarity (LPIPS) [53] to evaluate the supervised metrical scores between the reconstructed images and corresponding clean images. Meanwhile, we

**Table 1: Dataset description of five commonly utilized image deraining benchmark datasets.**

| Dataset | Rain800 | Rain1400 | Rain1200 | RainCityscapes | SPA-Data |
|---|---|---|---|---|---|
| Training | 700 | 12 600 | 12 000 | 9 432 | 28 500 |
| Testing | 100 | 1 400 | 1 200 | 1 188 | 1 000 |
| Testname | Test100 | Test1400 | Test1200 | Test1188 | Test1000 |

also utilize the contrastive language-image pre-training image quality assessment (CLIPIQA) [37], multi-dimension attention network image quality assessment (MANIQA) [46] and multi-scale image quality transformer (MUSIQ) [18] to evaluate the no-reference metrical scores of the derained images. The lower LPIPS score indicates better image quality, and vice versa for the others.

### 5.4 Comparisons with Existing Methods

To evaluate the effectiveness of our proposed approach on synthetic image deraining, we employ the combined training dataset to train the involved methods and conduct evaluation on the corresponding five testing datasets, respectively. As Table 2 depicted, our proposed method demonstrates substantial enhancements in performance metrics, surpassing existing methods in both reference-based and no-reference evaluations. To further illustrate the effectiveness of our model, we additionally compute the average quantitative metrics and offer intuitive comparisons among various methods across LPIPS and the three image naturalness assessment metrics shown in Figure 1. As illustrated, our model excels in average performance under supervised protocols and exhibits significantly enhanced naturalness compared to the other methods. Furthermore, we also provide several visual samples of the comparative methods in Figure 3. Specifically, although EGSDE [54] with a similar setup can ensure that the reconstructed images fall within the clean domain, which leads to severe artifacts and hallucinations due to the lack of strict constraints on details. Meanwhile, the visual results of the other approaches exhibit either notable residual degradation or substantial color distortions. In contrast, our model effectively removes rain streaks while maintaining consistent color consistency.

### 5.5 Ablation Studies

We conduct ablation experiments to evaluate the effectiveness of our model for unpaired photo-realistic image deraining, and all the experimental results are calculated by averaging the metrical scores across the five testing datasets.

*5.5.1 Individual Energy Functions.* We validate the effectiveness of our proposed dual-consistent energy function by changing the weighting parameters $\lambda_1$ and $\lambda_2$ to control the contribution level of the rain-relevant discarding and rain-irrelevance preserving energy functions in the overall performance. As shown in Figure 4, although the results generated without using our energy functions clearly fall within the clean domain, the textures of reconstructed images are inaccurate and there are many artifacts. This result intuitively proves the effectiveness of our dual-consistent energy function in constraining the domain and textures of reconstructed images. Meanwhile, Table 3 reports the quantitative comparisons

**Table 2: Quantitative comparisons of our proposed method and the other comparative algorithms on five synthetic testing datasets. The red and blue font indicate the best and second-best metrical scores, respectively. Our model achieves the best performance in both the reference-based and no-reference metrical indicators. Our work mainly focuses on achieving better photo-realistic deraining, thereby we provide the other metrical indicators in the supplementary materials to save space.**

| Method | Test100 | | | | Test1400 | | | | Test1200 | | | |
|---|---|---|---|---|---|---|---|---|---|---|---|---|
| | LPIPS | CLIPIQA | MUSIQ | MANIQA | LPIPS | CLIPIQA | MUSIQ | MANIQA | LPIPS | CLIPIQA | MUSIQ | MANIQA |
| CycleGAN [58] | 0.2908 | 0.3910 | 57.742 | 0.2294 | 0.2345 | 0.3774 | 59.068 | 0.2600 | 0.2600 | 0.3591 | 58.332 | 0.2359 |
| UAGGAN [1] | 0.4555 | 0.2736 | 54.166 | 0.2303 | 0.4566 | 0.3000 | 53.996 | 0.2512 | 0.4557 | 0.3012 | 55.511 | 0.2472 |
| CUT [25] | 0.3469 | 0.4099 | 57.041 | 0.2344 | 0.3445 | 0.4119 | 57.357 | 0.2513 | 0.3485 | 0.4025 | 56.991 | 0.2378 |
| DCLGAN [13] | 0.5387 | 0.2557 | 50.352 | 0.2110 | 0.5171 | 0.2636 | 52.773 | 0.2217 | 0.5486 | 0.2472 | 54.746 | 0.2301 |
| DerainCycleGAN [40] | 0.2117 | 0.5089 | 64.661 | 0.3744 | 0.2659 | 0.5004 | 62.626 | 0.4338 | 0.3867 | 0.3880 | 65.423 | 0.3395 |
| NLCL [47] | 0.4428 | 0.5358 | 57.934 | 0.3662 | 0.4000 | 0.5356 | 61.115 | 0.3972 | 0.4316 | 0.4777 | 60.056 | 0.3673 |
| DCD-GAN [6] | 0.3693 | 0.3223 | 54.857 | 0.2207 | 0.3623 | 0.3221 | 54.073 | 0.2333 | 0.3611 | 0.3188 | 54.799 | 0.2354 |
| LUD-VAE [55] | 0.4787 | 0.4900 | 54.024 | 0.2480 | 0.5280 | 0.4697 | 51.163 | 0.2613 | 0.6671 | 0.4201 | 49.645 | 0.2150 |
| EGSDE [54] | 0.2196 | 0.5427 | 60.105 | 0.3992 | 0.3184 | 0.5439 | 61.550 | 0.4091 | 0.2957 | 0.3864 | 66.025 | 0.3431 |
| UNSB [19] | 0.6457 | 0.2456 | 39.328 | 0.2076 | 0.6508 | 0.2820 | 40.882 | 0.2200 | 0.6475 | 0.2591 | 43.158 | 0.2301 |
| CLIP-LIT [21] | 0.2967 | 0.5724 | 62.978 | 0.3665 | 0.2124 | 0.5817 | 64.412 | 0.4221 | 0.2971 | 0.4792 | 64.507 | 0.3533 |
| **UPID-EDM (ours)** | 0.1996 | 0.5953 | 65.773 | 0.4176 | 0.1684 | 0.6404 | 66.164 | 0.4700 | 0.2108 | 0.4919 | 69.321 | 0.4011 |

| Method | Test1188 | | | | Test1000 | | | | Average | | | |
|---|---|---|---|---|---|---|---|---|---|---|---|---|
| | LPIPS | CLIPIQA | MUSIQ | MANIQA | LPIPS | CLIPIQA | MUSIQ | MANIQA | LPIPS | CLIPIQA | MUSIQ | MANIQA |
| CycleGAN [58] | 0.3611 | 0.1830 | 49.250 | 0.2141 | 0.6294 | 0.1599 | 44.388 | 0.2095 | 0.3552 | 0.2941 | 53.756 | 0.2298 |
| UAGGAN [1] | 0.3929 | 0.2763 | 45.774 | 0.2031 | 0.3934 | 0.2811 | 48.984 | 0.2543 | 0.4308 | 0.2864 | 51.686 | 0.2372 |
| CUT [25] | 0.2918 | 0.3102 | 52.324 | 0.1996 | 0.2567 | 0.2991 | 47.682 | 0.2491 | 0.3177 | 0.3667 | 54.279 | 0.2344 |
| DCLGAN [13] | 0.4186 | 0.2215 | 47.584 | 0.2053 | 0.4457 | 0.2329 | 47.394 | 0.2327 | 0.4937 | 0.2442 | 50.570 | 0.2202 |
| DerainCycleGAN [40] | 0.4549 | 0.3870 | 58.439 | 0.4180 | 0.2563 | 0.2252 | 49.268 | 0.2699 | 0.3151 | 0.4019 | 60.083 | 0.3671 |
| NLCL [47] | 0.3723 | 0.3738 | 53.014 | 0.4786 | 0.4475 | 0.2938 | 53.363 | 0.2951 | 0.4188 | 0.4433 | 57.096 | 0.3809 |
| DCD-GAN [6] | 0.2605 | 0.2502 | 49.967 | 0.2005 | 0.2430 | 0.2580 | 43.077 | 0.2412 | 0.3192 | 0.2943 | 51.355 | 0.2262 |
| LUD-VAE [55] | 0.8213 | 0.2334 | 47.532 | 0.1860 | 0.9337 | 0.3262 | 39.353 | 0.2142 | 0.6858 | 0.3879 | 48.343 | 0.2249 |
| EGSDE [54] | 0.3200 | 0.3522 | 50.954 | 0.4436 | 0.2600 | 0.3073 | 51.144 | 0.3042 | 0.2827 | 0.4265 | 57.956 | 0.3798 |
| UNSB [19] | 0.6294 | 0.2055 | 28.608 | 0.1533 | 0.5748 | 0.2943 | 47.297 | 0.2552 | 0.6296 | 0.2573 | 39.855 | 0.2132 |
| CLIP-LIT [21] | 0.2403 | 0.3139 | 58.298 | 0.3742 | 0.2669 | 0.2626 | 49.426 | 0.2732 | 0.2627 | 0.4420 | 59.924 | 0.3579 |
| **UPID-EDM (ours)** | 0.1503 | 0.4527 | 59.613 | 0.5315 | 0.1647 | 0.3428 | 57.085 | 0.3228 | 0.1788 | 0.5046 | 63.591 | 0.4286 |

of different weighting parameters, which illustrates that both our decomposed two energy functions present positive effect on the overall performance. In our experiments, we empirically determine that setting $\lambda_1$ to 73 and $\lambda_2$ to 0.72 strikes the desired balance between fidelity and naturalness.

**Table 3: Ablation experiments on the individual energy functions. Our energy function shows positive performance in improving the image fidelity and naturalness.**

| Weight | LPIPS | CLIPIQA | MUSIQ | MANIQA |
|---|---|---|---|---|
| $\lambda_1 = 50, \lambda_2 = 0$ | 0.4901 | 0.4298 | 52.047 | 0.2926 |
| $\lambda_1 = 100, \lambda_2 = 0$ | 0.5126 | 0.4445 | 53.900 | 0.3468 |
| $\lambda_1 = 200, \lambda_2 = 0$ | 0.5123 | 0.5136 | 60.321 | 0.3845 |
| $\lambda_1 = 0, \lambda_2 = 0.5$ | 0.4394 | 0.2970 | 46.996 | 0.2344 |
| $\lambda_1 = 0, \lambda_2 = 1$ | 0.3803 | 0.2793 | 45.234 | 0.2178 |
| $\lambda_1 = 0, \lambda_2 = 2$ | 0.3001 | 0.2802 | 43.077 | 0.1999 |
| $\lambda_1 = 73, \lambda_2 = 0.72$ | 0.1788 | 0.5046 | 63.591 | 0.4286 |

*5.5.2 Rain-relevance Discarding Energy Function.* We also compare our proposed rain-relevance discarding energy function with the texts and classifier. We substitute the learnable domain-representation prompts with the negative text $'rainy\ image'$ and the positive text $'clean\ image'$ in the texts. Additionally, we utilize a basic VGG16 network [29] to train a classifier for clarifying the rainy and clean images. As Table 4 reported, our LDP demonstrates the significant image fidelity and naturalness improvements over the simple texts and classifier. Therefore, our proposed energy function for discarding rain-relevance aids the pre-trained score-based diffusion model in eliminating the rain-relevant features.

*5.5.3 Rain-irrelevance Preserving Energy Function.* To further demonstrate the effectiveness of our proposed rain-irrelevance preserving energy function, we compare the performance of it with the commonly used cycle consistency constraint (CCC) [40] and the Gaussian-guided mean squared error (GMSE) [54]. As Table 5 depicted, our RPEF achieves the performance gain over both consistency-preserving functions, which demonstrates that our RPEF preserves

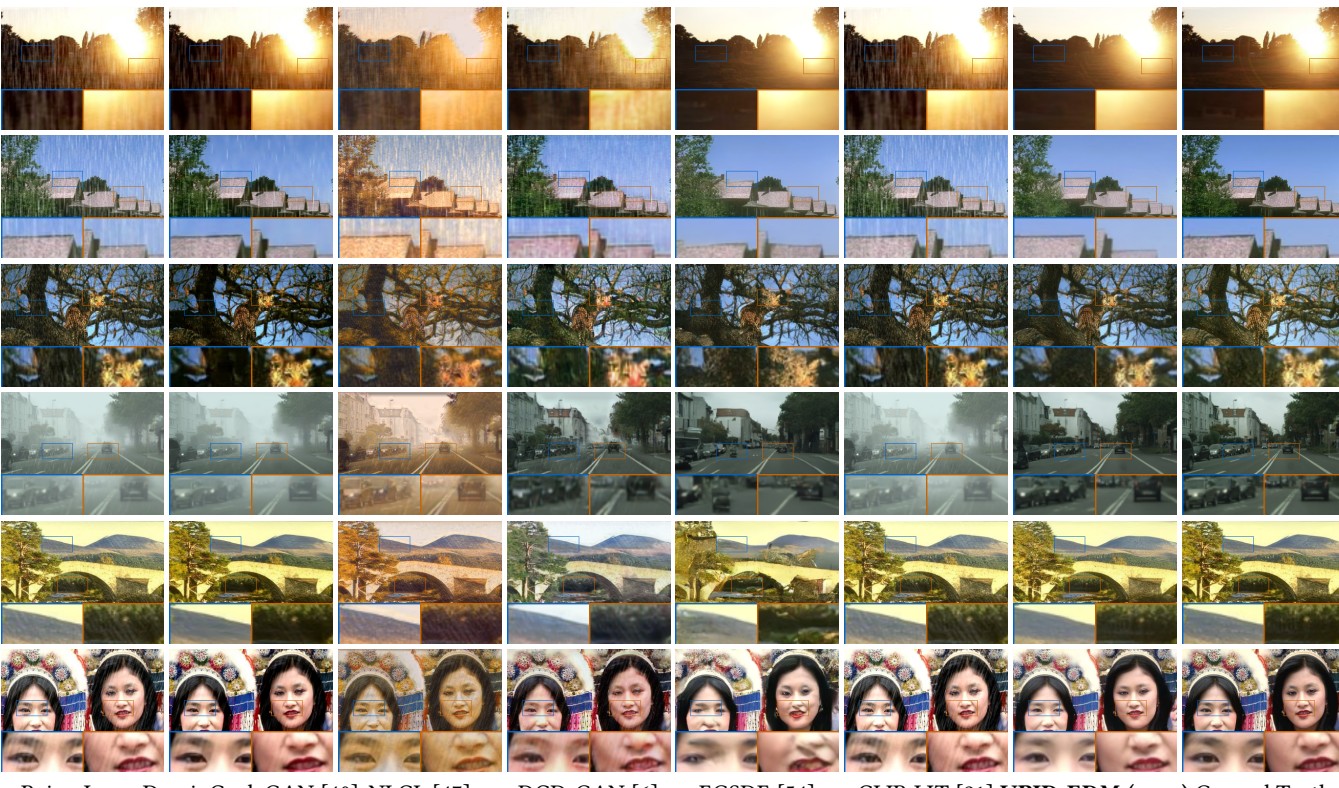

Rainy Image DerainCycleGAN [40] NLCL [47] DCD-GAN [6] EGSDE [54] CLIP-LIT [21] **UPID-EDM (ours)** Ground Truth

**Figure 3: Visual samples of the involved methods on synthetic rainy images. Our proposed approach completely eliminates the rain streaks and generates more photo-realistic derained images.**

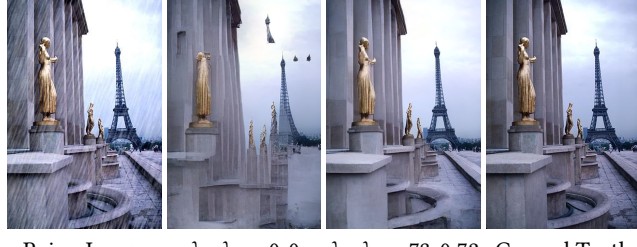

Rainy Image $\lambda_1, \lambda_2 = 0, 0$ $\lambda_1, \lambda_2 = 73, 0.72$ Ground Truth

**Figure 4: Intuitive comparisons of the effectiveness of the proposed dual-consistent energy function. Without our proposed dual-consistency energy function, the textures of reconstructed images are inaccurate and there are many artifacts.**

the rain-irrelevant features, and further improves the quality of the reconstructed images.

*5.5.4 Learnable Domain-representation Prompts.* Table 3 verifies the effectiveness of our proposed learnable domain-representation prompts. To clarify the motivation of our LDP, we additionally demonstrate the comparative latent similarities between various prompts and the provided rainy and clean images in Figure 5, where the texts similarities are compared with the given images, and the

**Table 4: Ablation experiments on the rain-relevance discarding energy function. Our learnable domain-representation prompts achieves better performance over the texts and classifier.**

| Component | LPIPS | CLIPIQA | MUSIQ | MANIQA |
|---|---|---|---|---|
| Texts | 0.2674 | 0.4828 | 57.644 | 0.4049 |
| Classifier | 0.2155 | 0.4420 | 59.724 | 0.4103 |
| **LDP** | 0.1788 | 0.5046 | 63.591 | 0.4286 |

**Table 5: Ablation experiments on the rain-irrelevance preserving energy function. Our RPEF achieves the better performance over the commonly used cycle consistency constraint [40] and the Gaussian-guided mean squared error [54].**

| Components | LPIPS | CLIPIQA | MUSIQ | MANIQA |
|---|---|---|---|---|
| CCC | 26.398 | 0.3466 | 59.440 | 0.3515 |
| GMSE | 0.1962 | 0.3790 | 60.325 | 0.3603 |
| **RPEF** | 0.1788 | 0.5046 | 63.591 | 0.4286 |

learnable prompts similarities are calculated across all the five testing datasets. In this figure, the negative prompts of given rainy

and clean images are $'rain\text{-}free\ image'$ and $'rain\text{-}degraded\ image'$, respectively. As illustrated, although the texts present similar meanings in depicting the provided images, they exhibit notable distinctions in the underlying context. This issue have also been demonstrated in [21]. Therefore, employing the texts to compute energy values in our rain-relevance discarding energy function would lead to instability in the reverse sampling procedure. In contrast, our LDP can classify rainy images and clean images more accurately.

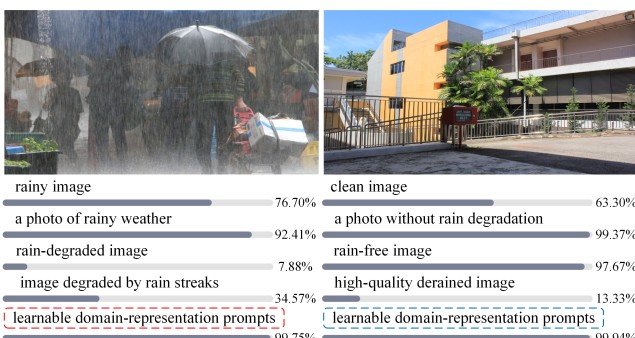

rainy image    76.70%

a photo of rainy weather    92.41%

rain-degraded image    7.88%

image degraded by rain streaks    34.57%

learnable domain-representation prompts    99.75%

clean image    63.30%

a photo without rain degradation    99.37%

rain-free image    97.67%

high-quality derained image    13.33%

learnable domain-representation prompts    99.94%

**Figure 5: Similarity comparisons of different prompts with the given images. Similar texts present significant distances in the latent space, while our learnable prompts achieve more accurate classification.**

*5.5.5 Starting Time.* We evaluate how the starting time parameter $T_s$ impacts the overall effectiveness of our proposed energy-informed diffusion model. The performance comparisons of our model with various initial time settings are detailed in Table 6. As depicted, our energy-informed diffusion model demonstrates comparable performance at both $T_s = 0.4T$ and $T_s = 0.6T$, and the under-performance at the other starting times. Therefore, we set $T_s = 0.4T$ to strike a balance between the reconstruction performance and inference time in our experiments.

**Table 6: Ablation experiments on the starting time. We set $T_s = 0.4T$ to achieve a balance between the performance and inference time in our experiments.**

| Initial Time | LPIPS | CLIPIQA | MUSIQ | MANIQA |
|---|---|---|---|---|
| $T_s = 0.2T$ | 0.2622 | 0.4860 | 62.156 | 0.4154 |
| $T_s = 0.4T$ | 0.1788 | 0.5046 | 63.591 | 0.4286 |
| $T_s = 0.5T$ | 0.2177 | 0.4964 | 61.164 | 0.4163 |
| $T_s = 0.6T$ | 0.1790 | 0.5050 | 63.580 | 0.4198 |
| $T_s = 0.8T$ | 0.3843 | 0.4751 | 60.492 | 0.3912 |

*5.5.6 Inputs of Energy Function.* Table 7 reports the performance of our approach when employing the $x_0$ and $x_t$ to calculate the energy values, where compute the energy values in the corresponding noise level results in an improved fidelity and naturalness performance. This proves that computing the energy values in the corresponding noise level ensures the model learn more accurate domain representation and image contents.

**Table 7: Ablation experiments on the inputs of energy function. Our proposed method achieves improved performance when employing both images in the same noise level to calculate the energy values.**

| Input | LPIPS | CLIPIQA | MUSIQ | MANIQA |
|---|---|---|---|---|
| $(y_t, x_0)$ | 0.3555 | 0.4690 | 60.123 | 0.4097 |
| $(y_t, x_t)$ | 0.1788 | 0.5046 | 63.591 | 0.4286 |

## 6 Limitations

Although our approach successfully eliminates the rain streaks without the requirements of paired rainy and cleans images, its major limitations arise from the significant computational resources required for generating the reconstructed images and the processing time for individual images surpasses that of the existing methods. Additionally, although our proposed approach is effective in removing the real-world rain degradation, which may inadvertently exhibit hallucinations (seen in Figure 6), leading to the degradation of image details and distortion of texture in several cases.

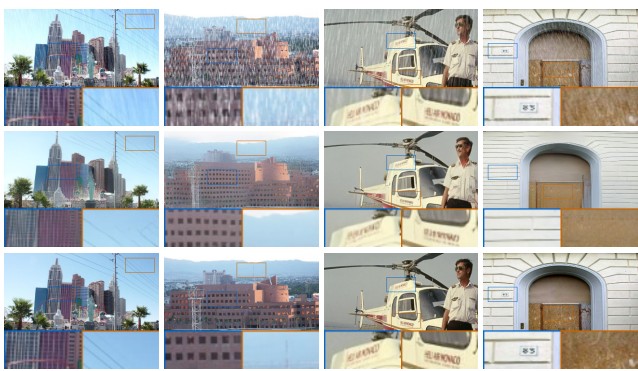

**Figure 6: Visual comparisons of the rainy, generated and clean images. Our model successfully eliminates the rain degradations, but generates several hallucinations.**

## 7 Conclusion

In this work, we introduce an innovative method for generating photo-realistic images without paired images. We delve into the rich visual-language priors within the contrastive language-image pre-training model, along with the capabilities of the score-based diffusion model in this context. Additionally, we showcase how our proposed dual-consistent energy function aids in filtering out the rain-relevant features while preserving the rain-irrelevant features during the reverse sampling process of a pre-trained diffusion model.

## Acknowledgments

This research was partially supported by the National Key R&D Program of China (Grant No. 2023YFB2504703), the Shaanxi International S&T Cooperation Program Project (Grant No. 2024GH-YBXM-24), the National Natural Science Foundation of China (Grant No. 52172379), and the Fundamental Research Funds for the Central Universities (Grant No. 300102242901).

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
