# OpenReview forum: "Unpaired Photo-realistic Image Deraining with Energy-informed Diffusion Model"
_acmmm.org/ACMMM/2024/Conference — MM2024 Oral_

### Official Review · Reviewer_zCRK · 2024-05-24

**Rating:** 3
**Confidence:** 3

**Summary:**

This work leverages contrastive language-image pre-training models and score-based diffusion models to generate photo-realistic images without paired images. The proposed dual-consistent energy function filters domain-related features while preserving domain-consistent features during reverse sampling.

**Strengths:**

1. The reported quantitative and qualitative results are good.
2. The proposed method uses rich priors from contrastive language-image pre-training to enhance image quality and diversity.
3. The proposed method combines score-based diffusion models, improving the stability and effectiveness of the generation process.

**Limitations:**

1. The authors mention in the paper that they are preserving the domain-consistent features (image content) while discarding the domain-related features (rain streaks). What is the basis for this assertion? Why are rain streaks categorized as domain-related features, and what exactly does "domain" refer to in this context?
2. The paper is hard to read and contains multiple instances of non-standard expressions. For example, in fig. 1, which dataset is used? Line 115, there are two dots. Line 178, "Specifically, We-->we ".
3. The experimental data presented in this paper need scrutiny. For example, the "DerainCycleGAN [17]" paper reports test results on the Rain800 dataset with PSNR 24.32 and SSIM 0.843, whereas this paper reports results of PSNR 24.257 and SSIM 0.7802. Given that this work utilizes more datasets for training compared to DerainCycleGAN, yet shows inferior results on the same test set, an explanation is required. Furthermore, Section 5.4 claims to test generalization ability, but according to my understanding, SPA-Data is a real dataset and was included in the training process. The authors should provide a clear explanation regarding this.

**Suitability:**

3

---

### Official Review · Reviewer_z5Rk · 2024-05-25

**Rating:** 5
**Confidence:** 2

**Summary:**

This paper introduces an energy-informed diffusion model for unpaired image deraining called UPID-EDM. The model utilizes the CLIP to distinguish between rainy and clean images and employs a dual-consistent energy function (DEF) to guide the reverse sampling process in a pre-trained diffusion model. This approach decomposes the energy function into two parts: one that discards domain-related features and another that preserves domain-consistent features. Sufficient experiments validate the effectiveness of the proposed model, demonstrating superior restoration quality and naturalness in the derained images.

**Strengths:**

This paper is well-organized. It effectively utilizes a dual-consistent energy function to separate and preserve domain-consistent features while discarding domain-related features, leading to more accurate and natural image reconstruction.

The authors present a novel view to handle this problem,  which is to train the domain-representation prompts based on the energy-based diffusion model to guide the deraining tasks during the denoising process.

**Limitations:**

[1]. In your Dual-consistent energy function, the energy function is divided into two parts: one is domain-consistent, and the other one is domain-related, which is similar to the EGSDE paper [51]. Meanwhile, Algorithm 1 is also similar to the algorithm presented in the EGSDE paper. Are there any improvements or differences in your algorithm and energy function?

[2]. Is there another way to choose the better value of $\lambda_1$ and $\lambda_2$? Could you show some qualitative results with different $\lambda_1$ and $\lambda_2$​​​?

[3]. The symbol in Line 316. is $q_{T_s|0} (y_{T_s}|x_0)$. The symbol in Line 345 is $q_{t|0}(.|.)$

**Suitability:**

2

---

### Official Review · Reviewer_hYkg · 2024-05-27

**Rating:** 5
**Confidence:** 2

**Summary:**

This paper proposes an Energy-informed Diffusion Model to solve the image-deraining problem. The authors delve into the intricate visual-language priors embedded within the contrastive language-image pre-training model (CLIP). The authors further employ the domain-relevance discarding energy function and the domain-consistency preserving energy function to direct the reverse sampling procedure of a pre-trained diffusion model, effectively removing the rain streaks while preserving the image contents. Extensive experiments demonstrate that our energy-informed model surpasses the existing unpaired learning approaches in terms of both supervised and no-reference metrics.

**Strengths:**

1, This paper is well-written.
2, This paper may be interesting.
3, The proposed UPID-EDM may be effective.

**Limitations:**

1, It would be better if the authors could state the difference between this paper and [A].

[A] EGSDE: Unpaired Image-to-Image Translation via Energy-Guided Stochastic Differential Equations. NIPS2022.

2, Missing some related references:
1) Dcsfn: Deep cross-scale fusion network for single image rain removal. ACM MM2020.
2) Joint self-attention and scale-aggregation for self-calibrated deraining network. ACM MM2020.
3) Online-updated high-order collaborative networks for single image deraining. AAAI2022.
4) PromptRestorer: A Prompting Image Restoration Method with Degradation Perception. NeurIPS2023.

**Suitability:**

2

---

### Meta-Review · Area_Chair_jJcJ · 2024-07-02

**Recommendation:** Accept (Oral)
**Confidence:** 5

**Metareview:**

This paper was reviewed by three experts in the field. The recommendations were mixed initially, but after authors’ feedback and discussion, the reviewers reached a consensus of acceptance, including Accept, Weak Accept, Borderline Accept. However, the reviewers still raise some valid concerns that authors did not fully resolve.

First, in the previous work EGSDE, the idea of domain specific and domain consistent features are also proposed. In the rebuttal, the authors only show quantitative comparison that shows the proposed method is better, but did not explain the reason and no additional experiment to verify why it is better. Second, experimental results for generalization are not unclear. It uses all five dataset, including real ones, and claims it can generalize to real images.

Still, AC agreed that this is an interesting idea and quantitative and qualitative results are good. Given that, the decision is to recommend the paper for acceptance to ACM Multimedia 2024.  We congratulate the authors on the acceptance of their paper!

Still, we recommended the authors to carefully read all reviewers’ final feedback, particularly the ones we mentioned above and revise the manuscript.